# A More Comprehensive Clinical and Laboratory Characterization of 61 Acute HIV Infection Patients in Southwest China

**DOI:** 10.3390/pathogens12010142

**Published:** 2023-01-14

**Authors:** Wu Shi, Mei Yang, Yinhao Wei, Zhuoyun Tang, Lan Luo, Jielun Deng, Chuanmin Tao

**Affiliations:** Department of Laboratory Medicine, West China Hospital, Sichuan University, Chengdu 610041, China

**Keywords:** acute HIV infection, characteristics, HIV Duo assay, rate of out-of-range tests, laboratory tests

## Abstract

Acute HIV infection (AHI), i.e., the early stage of HIV infection, plays an important role in immune system failure and HIV transmission, but most AHI patients are missed due to their non-specific symptoms. To facilitate the identification of patients with high AHI risk and reduction of missed diagnosis, we characterized 61 AHI patients in a Southwest China hospital with 4300 beds; specifically, we characterized their general clinical characteristics, evolution in results of a novel HIV screening assay called Elecsys^®^ HIV Duo, and by programming, we analyzed the ability of all routine laboratory tests (e.g., routine blood analysis) to identify AHI patients. Among 61 AHI patients, 85.2% were male and the median age was 42 (interquartile range, 25–62) years. A total of 61.9% of patients visit the emergency department first during AHI. Clinical presentation of AHI patients included fever, fatigue, chills, rash, and various respiratory, digestive, and nervous system symptoms. Two of three results from Elecsys^®^ HIV Duo show clear evolution trends: HIV P24 antigen decreased while HIV antibody increased in consecutive samples of nearly all patients. High fluorescence lymphocytes have a very high positive likelihood ratio (LR+) of 10.33 and a relatively high “rate of out-of-range tests” of 56.8% (21 in 37 patients who received this test had a result outside the reference range). In addition, we identified more than ten tests with LR+ greater than two. In summary, the emergency department is important for AHI screening. The evolution of HIV P24 Ag and HIV Ab and those laboratory tests with a high “rate of out-of-range tests” or high LR+ may aid the AHI identification and missed diagnosis reduction.

## 1. Introduction

Human immunodeficiency virus (HIV) infection remains a serious public health problem, causing 37.7 (30.2–45.1) million infections and 0.68 (0.48–1.0) million deaths in 2020 [1] After infection with HIV, HIV RNA, HIV P24 antigen (Ag), and HIV antibody (Ab) would appear sequentially in serum. Acute HIV infection (AHI) refers to the period that begins with the appearance of detectable HIV RNA in serum and ends with the conversation of HIV western blot (WB) from positive without a P31 band to positive with a P31 band [2].

At the early stage of AHI, before the establishment of HIV-specific immune response, the HIV replicates exponentially, and HIV genome is massively integrated, leading to substantial depletion of CD4+ T cells and extensive establishment of viral reservoirs. In addition, due to the higher viral load and greater infectivity of individual virions [3,4], approximately 10–50% of HIV transmissions are attributed to individuals infected recently [5]. Thus, early initiation of antiretroviral therapy (ART) during AHI can potentially reduce the immune system compromise [6,7] and decrease the risk of onward transmission [5]. 

Medical visits for symptoms are frequent in AHI patients [8], but few of them would receive HIV screening tests during the visit [9] because their symptoms are generally non-specific [10,11]. Furthermore, the immature anti-HIV immunity of AHI patients cannot enable a positive result of the mainstream HIV antibody confirmatory test and would result in a 2–4 week follow-up with a high rate of loss to follow-up [9]. For these reasons, most AHI patients are missed. In China, late diagnosis is similarly common among HIV-infected patients [12].

Therefore, it is important to describe the characteristics of AHI, so that patients at high risk of AHI could be identified based on these characteristics, and by concentrating limited testing resources on them, more AHI patients could be diagnosed without excessive costs [5,13].

Previous studies on this topic focused primarily on the symptoms of AHI, however, Risk-score algorithms based on symptoms showed a limited effect on identifying AHI patients. This article provides a comprehensive analysis of the laboratory characteristics of patients with AHI, which have been described in a very limited way previously. Specifically, we analyzed the dynamic change in results of a novel HIV screening assay called Elecsys^®^ HIV Duo, and by programming, we analyzed the ability of all routine laboratory tests (e.g., routine blood analysis) to identify AHI patients.

## 2. Materials and Methods

### 2.1. Inclusion Criteria and Patients

In this paper, AHI was defined as the first five stages of the Fiebig staging system, which is most widely used. Based on the definition, patients meeting each of the following inclusion criteria were enrolled in this study: (1) HIV WB results changed from negative/indeterminate to positive; (2) negative or indeterminate WB results plus detectable HIV-1 RNA; (3) positive WB result without p31 band but with the typical acute retroviral syndrome (ARS) [10]. These enrollment pathways for AHI patients are shown in Figure 1.

According to these inclusion criteria, a total of 61 eligible patients were selected for further analysis from those who received HIV screening in West China Hospital between January 2017 and October 2021. This study was approved by the ethics committee of the West China Hospital (approval number: 2020742).

### 2.2. Data Collection

Data on demographics, first medical visit experience, signs, and symptoms were extracted from the hospital information system (HIS). All laboratory tests for each patient were exported from the laboratory information system (LIS). The exposure history investigated by the Chinese Center for Disease Control and Prevention (CCDC) was queried. Unfortunately, the CCDC only allowed us to query patients who were reported to CCDC at our hospital; for some patients who were referred to other hospitals and reported at those hospitals, their exposure history was inaccessible to us and was recorded as “unknown”.

### 2.3. Elecsys^®^ HIV Duo, Supplementary Test and Staging of the Infection

Being different from classical fourth-generation (4G) HIV screening assays, which provide only combo test results of HIV P24 Ag and HIV Ab, Elecsys^®^ HIV Duo lists independent results of them in addition to the combo test results. From January 2017 to March 2020, 35 patients were screened with a conventional 4G assay: Elecsys^®^ HIV combi PT (Roche Diagnostics, Mannheim, Germany). Since April 2020, the HIV Duo assay was introduced into our clinical laboratory, and 26 patients were tested with this assay. The unit of the result produced by both assays is the ratio of signal to cut-off and is designated as “cut-off index (COI)”.

After the positive HIV screening test, patients would retake a blood sample for the WB test (Western Blot HIV blot 2.2, MP Diagnostics, Singapore). On each WB sample, the 4G screening assay would also be performed. On each screening and WB sample, a third-generation (3G) assay (AntiHIV Colloidal gold kit (Zhuhai Livzon Diagnostics Inc., Zhuhai, China)) would also be performed until the 3G assay result turned positive. Moreover, a significant proportion of patients did not accept the fact that they were infected until obtaining a positive WB result, and these patients would take multiple screening and WB samples until the WB result progressed to positive.

As the definition mentioned above, AHI can be further divided into five stages of the Fiebig system, and each stage was indicated by Roman numerals I–V, separately. These stages are determined based on the results of the 4G assay, 3G assay, and WB test. Therefore, for each patient awaiting WB results progression, their multiple samples were in different Fiebig stages. Based on these samples, a comparison of the screening test results between different stages was conducted. In addition, based on multiple screening and WB samples of each patient, evolution trends of screening test results were analyzed.

### 2.4. The Ability of All Routine Laboratory Tests to Identify AHI Patients

Among HIV-negative individuals from the same department on the same day as the AHI patients, we matched each AHI patient with 3 control patients by random sampling. Then we calculated the “rates of out-of-range tests” (number of tests outside the reference range/total number of tests) of each laboratory test in the AHI group and HIV-negative group and compared the statistical difference in “rates of out-of-range tests” between the two groups, and finally we calculated the positive likelihood ratio (LR+) for identifying AHI for these laboratory tests.

The mechanism to calculate the “rate of out-of-range test” is as follows: For test records with a result outside the corresponding reference range, LIS would place a marker like “↑” on its qualitative field. In contrast, the qualitative field will be null when the result was within the reference range. We calculated the number of records with a marker and the total number of records, then divide the two to obtain the “rate of out-of-range test”.

The same patient may undergo the same laboratory test before HIV infection, after receiving ART, or after the infection progress to the chronic stage. In these scenarios, the test result is unrelated to AHI. Therefore, for each laboratory test, the time interval from that test to the same patient’s first positive HIV screening test was calculated. Only if the absolute value of the time difference was ≤3 days, the test results were included for subsequent analysis.

This analysis was conducted with a programming language: Python (version 3.8.10). The complete code and some auxiliary datasets for “rate of out-of-range test” calculation have been uploaded to the Github (https://github.com/shiwu-labmed/abnormality-rate-calculation (accessed on 24 October 2022)), a example dataset of laboratory tests results was provided for code validation (see Appendix A). The calculated “rate of out-of-range test” were sampled for manual verification.

### 2.5. Statistical Analysis

Statistical analyses were conducted in R version 4.1.3. Qualitative data were presented as frequencies and percentages. Quantitative data were presented as median (interquartile range (IQR)). The Chi-square test were used to compare frequencies. The Kruskal–Wallis *H* test followed by a post-hoc Dunn’s test was conducted to test the differences across multiple independent samples, the Wilcoxon sign rank test was conducted to test the differences between two independent samples, the Kendalls’ Tau-b correlation was used to analyze the correlation between ordinal variables and continuous variables. Statistical tests were 2-sided, and *p* < 0.05 was considered as the difference with statistical significance.

## 3. Results

### 3.1. Demographics, Exposure History and First Medical Visit Experience

Of the 61 patients who met the inclusion criteria, 52 (85.2%) were male, the median age at diagnosis was 42 years, with a wide IQR of 25–62 years, and median age and IQR for males and females were 43 (24,63) and 51 (49,56), respectively.

During the course of AHI prior to inclusion in the study, four patients had a history of visits to other hospitals and five had a history of visits to our hospital, and no one was diagnosed with HIV infection during these visits.

Excluding the 19 individuals with unknown exposure history, the remaining patients were mainly heterosexually exposed patients (61.9%, 26/42), followed by men having sex with men (MSM) (33.3%, 14/42), and patients with other exposure histories (4.8%, 2/42).

Fourteen patients were tested for HIV viral load within 3 days before and after their HIV screening, with a median (IQR) of 5.75 × 10^6^ (8.5 × 10^5^, 1 × 10^7^) copies/mL, and 26 patients were tested for CD4+ T-cell count with a median (IQR) of 214.0 (117.8–292.5) cells/µL.

Please see Table 1 for details.

The age distribution of men has two peaks at the 16–26 age group and the 57–66 age group, and the composition of exposure history differs between the two peaks, with the former peak consisting mainly of MSM and the latter mainly of commercial heterosexually exposed patients. Specifically, excluding those with unknown exposure history, 82% (14/17) of male patients under 36 were MSM, while 77.8% (14/18) of male patients over 37 had a commercial heterosexual exposure history. In contrast to men, the age distribution of women has a single peak consisting of only one type of exposure history: non-commercial heterosexual exposure. See Figure 2A for detailed distribution of exposure history across age and gender.

The first visit departments during the episode of AHI were dominated by emergency department (62.3%, 38/61), followed by infectious disease center (9.8%, 6/61), convenience clinic (8.2%, 5/61), department of general surgery (6.5%, 4/61), Mental Health Center (3.2%, 2/61), and other departments (9.8%, 6/61) (Figure 2B). The first medical visit occurred primarily in summer and autumn, with a peak frequency in August (Figure 2C).

### 3.2. Frequency of Symptoms and Signs

The top three symptoms/signs most frequently reported were fever (50.82%, 31/61), dizziness and/or headache (37.70%, 23/61), and cough and/or expectoration (26.23%, 16/61). The nervous system was the most common organ system to exhibit symptoms and signs (44.26%, 27/61), followed by digestive system (37.70%, 23/61) and respiratory system (34.43%, 21/61). Moreover, 19.67% of patients exhibited no symptoms and were diagnosed by preoperative screening/voluntary counseling and testing (VCT). The complete relative frequency of symptoms and signs is shown in Table 2.

### 3.3. Analysis of the Results of Elecsys^®^ HIV Duo

Despite the presence of outliers, the combo test results of HIV P24 Ag and HIV Ab (i.e., HIVCOM and HIV Duo) showed a pronounced initial decline and then a rising trend as the Fiebig stage advanced (Figure 3A,B). As shown in Figure 3A, patients were in different Fiebig stages at presentation; consequently, the combo test results may increase or decrease after their presentation, and no regular pattern could be observed for the evolution of the combo test results (Figure 3C). However, with regard to independent results of HIV P24 Ag and HIV Ab, which had statistically significant positive and negative correlations with Fiebig stage, respectively (Figure 3D,F), the situation was much different; a sample-by-sample reduction of HIV P24 Ag (Figure 3E) and a sample-by-sample elevation of HIV Ab (Figure 3G) could be clearly observed in nearly all patients.

No patient was found to be in the Fiebig stage I, because this stage was defined as a period with the detectability of HIV RNA and the undetectability of any other HIV biomarker. This implies that patients in this stage could not be identified by our screening assay, which could not detect HIV RNA.

### 3.4. The Ability of All Routine Laboratory Tests to Identify AHI Patients

To avoid an overly lengthy table, only laboratory tests that differed significantly between the AHI and HIV-negative groups are shown (Table 3). A complete table was provided in the Appendix A to list all laboratory tests that were out of the reference range in more than five patients.

Overall, these tests that differed significantly between the two groups fall into five main groups: analysis of T-cell subsets (e.g., CD4+ T-cell counts) and inflammatory markers (e.g., procalcitonin), routine urine test (e.g., urine protein), blood routine analysis (e.g., absolute lymphocyte count), and blood biochemical analysis (e.g., lipoprotein cholesterol).

The highest “rate of out-of-range test” was observed in procalcitonin—of 32 patients receiving this test, 97% (30/31) had a result outside the reference range, but its LR+ was only 1.25. In contrast, high fluorescence lymphocytes have a moderate “rate of out-of-range tests” (56.8%, 21/37) and the highest LR+ (10.33).

## 4. Discussion

From 2009 to 2020, the mode of HIV transmission has changed extensively, with heterosexual and homosexual transmission increasing from 48.3% and 9.1% to 74.2% and 23.3%, and injecting drug transmission decreasing from 25.2% to <2.5% [14]. Our data were consistent with it; patients with sexual exposure history accounted for 95.2% (40/42) after excluding those with unknown exposure history. 

The overall age distribution of HIV infection in China varied across studies. Qiao’s study [15] showed a unimodal age distribution from 2004–2016, with peaks located at 20–30 years old, while Liu’s study [16] showed a bimodal age distribution of HIV infection from 2004–2014, with two peaks located at 25–34 and 55–64 years old, respectively. Our results are similar to those of Liu’s study.

Furthermore, we found that the composition of exposure history varied by age and sex; younger men were mainly MSM, and most older men had commercial sexual exposure history, whereas in women it was all non-commercial heterosexual. This may explain the bimodal age distribution of men and the higher incidence in men than in women.

A considerable number of studies show that screening for AHI in the emergency department has a high yield [17] and can even be comparable to the high-prevalence populations of STI clinic attendees [18]. Our results support their conclusion from another perspective, with 62.3% of AHI patients visiting the emergency room first. As for the distribution of the time of the first visit, it may be because there are more influenza cases in the winter and spring and therefore more diseases with similar symptoms to mislead patients and doctors.

Most of the highly frequent symptoms in this study, such as fever, dizziness/headache, and cough/expectoration, are consistent with those reported in other populations [10,19,20]. However, relative to prior research, the lymphadenopathy was infrequent in this study, although a similar report among Asian populations exists [21]. There is another possible explanation: the enlargement of lymph nodes was minimal [19], and was therefore easly to be neglected. In contrast, dyspnea-related symptoms have been rarely reported before, except in an earlier prospective study [22] and several case reports [11,23,24]. This may be partly attributed to the fact that the descriptions of dyspnea-related symptoms are too diverse, making it difficult to recognize that these descriptions can be grouped together, even though they have similar meanings and rarely occur alone in one patient.

The evolution in Elecsys^®^ HIV Duo results can be explained theoretically with the natural course of AHI [2,25]. After exposure and transmission of HIV, the serum levels of HIV P24 Ag increase with the exponential viral replication, and once the HIV P24 Ag concentration is high enough to be detected, the infection has progressed into Fiebig stage II. Then, with the activation of immunity and the following inhibition of HIV replication, HIV P24 Ag levels decrease and HIV Ab increases in serum. Once the HIV Ab was detected in serum, the infection reaches the Fiebig stage III. Therefore, HIV P24 Ag decreases after stage II and HIV Ab increases after stage III. Based on the premise that the earliest stage we can find was Fiebig stage II, it was not surprising to find an increasing trend in HIV P24 Ag and a decreasing trend in HIV Ab. Furthermore, during stages II and III, the reduction in the combo test results was due to the decreasing HIV P24 Ag, and the elevation during stages IV and V was due to the increasing HIV Ab.

Another interesting finding was the detectable HIV P24 Ag after Fiebig stages III. Theoretically, the HIV Ab present after stage III could adversely affect antigen detection because the HIV Ab could close the antigen binding site, which is necessary for HIV P24 Ag detection. A possible explanation might be that HIV Duo has specialized antigen dissociation reagents, which could release the closed sites, so that in our study, the p24 antigen could still be detected in most patients at Fiebig stage III (100%, 3/3) and IV (81%, 22/27). In clinical practice, this detectability played an important role in AHI identification, because the HIV P24 Ag is a biomarker of AHI.

Today’s mainstream HIV antibody confirmatory test, that is, WB, is negative at Fiebig stages II and III and undetermined at Fiebig stage IV [2]. The former situation leads to missed diagnosis, while the latter requires 2–4 weeks of follow-up, but this follow-up has a high rate of loss [9]. Therefore, AHI patients may be missed even if they have been screened for HIV. However, because of the clear characteristics of HIV Duo results in AHI patients, we can enhance their exposure history inquiry and follow-up in a more targeted manner to reduce missed diagnoses resulting from negative or undetermined WB. Indeed, since introducing HIV Duo, the average number of AHI diagnosed per month in our laboratory has increased to 1.44 cases (26 cases/18 months), compared to 0.875 cases (35/40) per month previously.

By analyzing all laboratory tests that AHI patients had received, we identified more than ten tests with LR+ greater than two. Among them, high fluorescence lymphocytes had the highest LR+ (10.33), which was much higher than all other tests, so this test may identify AHI well. The test also had a “rate of out-of-range tests” of 56.8% in AHI patients, which means that this test could identify nearly 60% of AHI patients. Most of the high-fluorescence lymphocytes in blood represent activated B cells and plasma cells, and their results are closely correlated with atypical lymphocytes obtained by manual microscopic count [26], but their association with AHI has not been previously reported.

Other tests with a high ability to identify AHI include lactate dehydrogenase, hydroxybutyrate dehydrogenase, and high-density lipoprotein cholesterol. These tests have relatively high LR+, as well as the ability to identify about 70% of patients with AHI, and therefore have potential value in the identification of AHI patients. Among them, the association between lactate dehydrogenase and AHI has been previously reported infrequently and is only found in a few case reports [27,28]. Hydroxybutyrate dehydrogenase, a subclass of lactate dehydrogenase, has not been previously reported to be associated with AHI. High-density lipoprotein cholesterol, on the other hand, has been suggested to be associated with cardiovascular disease in AHI patients in previous studies [29,30].

An interesting finding is that urine protein and urine epithelial cells are associated with AHI. No similar report has been previously seen, and in fact, urine specimens have received little attention in AHI-related studies. Unfortunately, however, the LR+ for urine protein was only 1.48 and urine epithelial cells could only identify a relatively small number of patients with AHI (44%, 11/25), so their role in the identification of patients with AHI is limited.

These tests found in our study can be used to develop risk-scoring algorithms to identify patients at high AHI risk. As not all patients would receive these laboratory tests, laboratory-based algorithms cannot be applied to all patients like the commonly used symptom-based algorithms. Fortunately, most of tests found in our study were from routine clinical tests, which most patients would receive at the time of their visit, so the laboratory-based algorithm can still be applied to most people.

There are some limitations in this study. First, the original data come from a single hospital and are not fully representative or applicable to the entire local or even national population. We hope to expand the number and diversity of samples by collecting data in different regions. Second, for unknown reasons, some tests would not be marked with any abnormal flag, even if they were outside the reference range, these tests were neglected by the “rate of out-of-range tests” analysis.

## 5. Conclusions

In summary, the composition of exposure history varied in different ages. The emergency department was an important setting for AHI screening. The HIV Duo assay may reduce the loss to follow-up of AHI patients thanks to the downward trend of its HIV P24 Ag results, the upward trend of its HIV Ab results, and its capacity to detect HIV P24 Ag from Fiebig II–V stage. High fluorescence lymphocytes is the most valuable test in identifying patients with AHI. Tests with LR+ >2 such as lactate dehydrogenase, hydroxybutyrate dehydrogenase, and high-density lipoprotein cholesterol are also valuable. Building a risk-scoring algorithm based on these tests may allow for better identification of AHI, but an algorithm built on prospective data may be more valuable, so a specific algorithm was not built in this study, and it is hoped that future prospective studies will give more attention to these tests.

## Figures and Tables

**Figure 1 pathogens-12-00142-f001:**
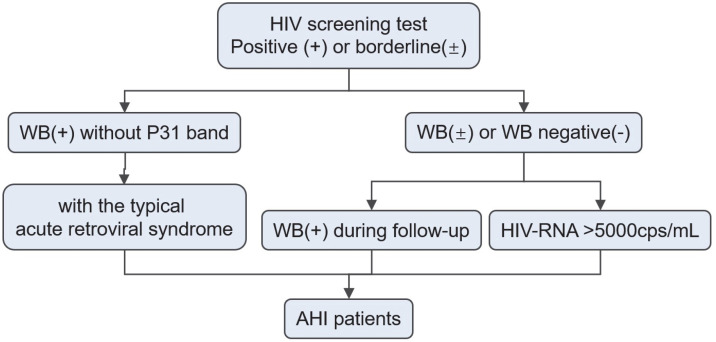
Enrollment pathways for AHI patients. Abbreviation: AHI, Acute HIV Infection; WB, Western Blot.

**Figure 2 pathogens-12-00142-f002:**
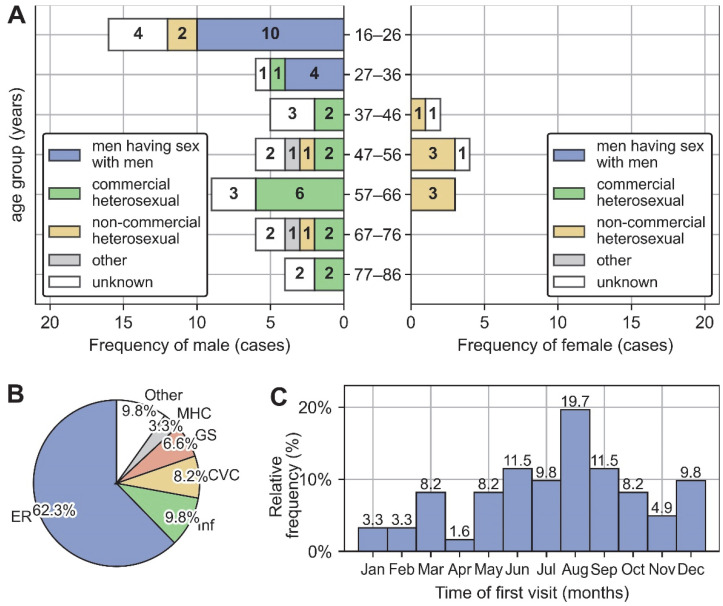
General clinical characteristics of 61 AHI patients. (**A**) Distribution of exposure history across age and gender. (**B**) Department composition and time composition (**C**) of the first medical visit during the episode of AHI. Abbreviation: ER, Emergency Room; Inf, Infectious Disease Center; CVC, Convenience Clinic; GS, Department of General Surgery; MHC, Mental Health Center.

**Figure 3 pathogens-12-00142-f003:**
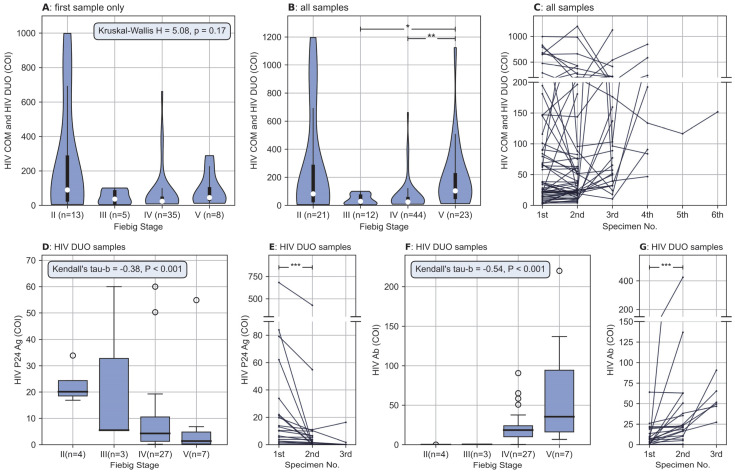
Association between results of HIV screening assay and Fiebig stages, and sample-by-sample evolution of HIV screening assay in each patient. (**A**) Different distribution of the HIV COM and HIV DUO between Fiebig stages, for each patient, only the first sample with sufficient information to enable Fiebig staging was included for analysis, differences were not statistically significant (H = 5.085, *p* = 0.166). Data are shown as violin plots, the violins indicate Gaussian kernel density estimates, the white dots denote the median, thick bars represent IQR, and thin lines extend 1.5 times the IQR. (**B**) Similar to (**A**), but all samples with different Fiebig stages were included for each patient, the differences were statistically significant (H = 16.538, *p* < 0.01). The HIV COM and HIV DUO were significantly higher at the Fiebig V stage than at the III and IV stages (*p* < 0.05 with Bonferroni correction). (**C**) Sample-by-sample evolution of HIV COM and HIV DUO, no regular pattern of evolution was observed. (**D**) Statistically significant negative correlations between HIV P24 Ag and Fiebig stage (Kendall’s tau-b = −0.378, *p* < 0.01). When plotting only, results above 60 were adjusted to 60 so that the box would not be overly compressed. Data are shown as box plots, the thick horizontal line in the box represents the median, the lower and upper boundary of the box represent the IQR, the whiskers extend 1.5 times the IQR, and the white dots represent outliers (outliers are values outside the range of the whisker line) (**E**) sample-by-sample reduction of HIV P24 Ag. For each patient, the HIV P24 Ag was significantly higher in the first sample than in the second sample (Z = −3.845, *p* < 0.01). (**F**) Statistically significant positive correlations between HIV Ab and Fiebig stage (Kendall’s tau-b = 0.537, *p* < 0.01). Similar to (**E**), results above 220 were adjusted to 220. (**G**) Sample-by-sample elevation of HIV Ab. For each patient, the HIV Ab was significantly lower in the first sample than in the second sample (Z = −3.733, *p* < 0.01). * denotes *p*< 0.05, ** *p*< 0.01 and *** denotes *p* < 0.001. Abbreviation: HIV COM, the combo test results of HIV P24 Ag and HIV Ab provided by HIV combi PT assay; HIV DUO, the combo test results provided by HIV Duo assay; COI, cut-off index.

**Table 1 pathogens-12-00142-t001:** Demographic and epidemiological characteristics of AHI patients.

	Acute HIV Infection
Male, percentages	85.2% (52/61)
Age, median (Q1, Q3)	42 (25, 62)
Age of male, median (Q1, Q3)	43 (24, 63)
Age of female, median (Q1, Q3)	51 (49, 56)
Exposure history, percentages (Unknown exposure were excluded)	
Heterosexual exposure	61.9% (26/42)
Men having sex with men	33.3% (14/42)
Other	4.8% (2/42)
viral load, median (Q1, Q3) [*n*] ^a^	5.75 × 10^6^ (8.5 × 10^5^, 1 × 10^7^) copies/mL [*n* = 14]
CD4+ cells counts, median (Q1, Q3) [*n*] ^a^	214.0 (117.8, 292.5) cells/µL [*n* = 26]

^a^ Only a subset of patients received this laboratory test within 3 days before and after screening; the letter n indicates the number of patients who received the test. Abbreviations: Q1, 25th quantile; Q3, 75th quantile.

**Table 2 pathogens-12-00142-t002:** The relative frequency of symptoms and signs reported during AHI.

Symptoms and Signs	Relative Frequency (Frequency/Count)	Total Relative Frequency (Frequency/Count) by Organ System
Fever	50.82% (31/61)	
Fatigue	24.59% (15/61)	
Chill	16.39% (10/61)	
Rash	9.84% (6/61)	
Cough and/or expectoration	26.23% (16/61)	respiratory system 34.43% (21/61)
Dyspnea related symptoms ^a^	14.75% (9/61)	
Pharyngalgia	11.48% (7/61)	
Hemoptysis	4.92% (3/61)	
Diarrhea	16.39% (10/61)	digestive system 37.70% (23/61)
Abdominal pain and discomfort	14.75% (9/61)	
Abdominal distension	8.20% (5/61)	
Vomiting	9.84% (6/61)	
Other digestive symptoms ^b^	16.39% (10/61)	
Dizziness and/or headache	37.70% (23/61)	nervous system 44.26% (27/61)
Other nervous symptoms ^c^	14.75% (9/61)	
Other symptoms ^d^	34.43% (21/61)	
Asymptomatic	19.67% (12/61)	

^a^ including a variety of descriptions, such as shortness of breath, chest tightness, tightness in breathing, breathlessness, and dyspnea. ^b^ including heartburn, acid regurgitation, retching, nausea, inappetence, hiccups, and hematochezia ^c^ including visual impairment, slurred speech, occlusal asymmetry, twitches, photophobia, sleepiness, and numbness. ^d^ including myalgia, bodily pain, mouth ulcers, hematuria, dry eye, dry mouth, thrombocytopenia, chest pain, palpitations, herpes zoster, lymphadenopathy, oral bleeding, discomfort, and night sweats.

**Table 3 pathogens-12-00142-t003:** The ability of all routine laboratory tests to identify AHI patients.

Test Type	Test Name	Rate of Out-of-Range Test (AHI)	Median [IQR] (AHI)	Rate of Out-of-Range Test (HIV-)	Median [IQR] (HIV-)	LR+	χ2	*p*	Reference Range	Unit *
T-cell subsets analysis	CD4+ T-cell counts	96.2% (25/26)	214.0 [117.8–292.5]	60.0% (6/10)	386.5 [285.8–592.5]	1.6	5.16	0.0231	471–1220	cell/µL
	CD4 cell percentage	88.9% (32/36)	11.6 [7.5–25.6]	51.7% (15/29)	37.4 [27.6–45.7]	1.72	9.3	0.0023	33.19–47.85	%
	CD4/CD8 ratio	83.3% (30/36)	0.2 [0.1–0.8]	37.9% (11/29)	1.7 [1.0–2.1]	2.2	12.33	0.0004	0.97–2.31	N/A
Inflammatory marker	Procalcitonin	96.8% (30/31)	0.2 [0.1–0.2]	77.4% (48/62)	0.1 [0.1–0.2]	1.25	4.38	0.0363	<0.046	ng/mL
Routine urine test	Urine Protein	74.2% (23/31)		50.0% (41/82)		1.48	4.42	0.0355	Negative	N/A
	Epithelial cells	44.0% (11/25)	4.0 [2.0–12.0]	19.5% (16/82)	4.0 [2.0–11.8]	2.26	4.86	0.0275	0–20	/µL
Routine blood analysis	High fluorescence lymphocytes	56.8% (21/37)	2.0 [0.9–3.2]	5.5% (6/109)	0.1 [0.0–0.2]	10.33	44.8	0	<1	%
	Eosinophil percentage	56.5% (26/46)	0.2 [0.0–1.0]	36.2% (54/149)	1.0 [0.2–2.7]	1.56	5.17	0.023	0.4–8.0	%
	Absolute Eosinophil Value	53.2% (25/47)	0.0 [0.0–0.0]	26.4% (39/148)	0.1 [0.0–0.2]	2.02	10.47	0.0012	0.02–0.52	10^9^/L
	Monocyte percentage	35.8% (19/53)	8.2 [6.3–11.0]	16.6% (25/151)	6.1 [4.6–8.3]	2.16	7.53	0.0061	3–10	%
	Mean Erythrocyte HGB Concentration	28.3% (15/53)	334.0 [323.0–349.0]	11.2% (17/152)	332.5 [323.0–339.2]	2.53	7.49	0.0062	316–354	g/L
	Mean red blood cell HGB content	24.5% (13/53)	30.6 [29.1–31.7]	11.2% (17/152)	30.4 [28.9–31.4]	2.19	4.58	0.0323	27–34	pg
Blood biochemical analysis	High-density lipoprotein cholesterol	73.9% (34/46)	0.8 [0.7–0.9]	32.2% (46/143)	1.1 [0.8–1.5]	2.3	23.17	0	> 0.90	mmol/L
	Lactate dehydrogenase	71.7% (33/46)	310.0 [231.5–454.0]	30.7% (43/140)	194.0 [155.8–241.0]	2.34	22.45	0	120–250	IU/L
	Hydroxybutyrate dehydrogenase	69.6% (32/46)	225.5 [170.0–339.5]	28.6% (40/140)	150.0 [120.5–188.5]	2.43	22.83	0	72–182	IU/L
	Sodium	68.2% (30/44)	134.8 [132.4–137.4]	30.7% (42/137)	139.1 [136.4–141.0]	2.22	18.04	0	137.0–147.0	mmol/L
	Albumin	65.3% (32/49)	38.3 [34.8–41.7]	44.7% (67/150)	41.0 [36.3–45.8]	1.46	5.5	0.0191	40.0–55.0	g/L
	Calcium	61.9% (26/42)	2.1 [2.0–2.1]	27.3% (36/132)	2.2 [2.1–2.3]	2.27	15.19	0.0001	2.11–2.52	mmol/L
	Aspartate Transaminase	53.1% (26/49)	41.0 [27.0–90.0]	20.7% (31/150)	22.0 [17.0–34.0]	2.57	17.41	0	< 35	IU/L
	Serum beta- hydroxybutyrate	50.0% (21/42)	0.3 [0.2–0.4]	28.1% (38/135)	0.1 [0.1–0.3]	1.78	5.94	0.0148	0.02–0.27	mmol/L
	Inorganic phosphorus	47.6% (20/42)	0.9 [0.7–1.0]	28.0% (37/132)	1.1 [0.9–1.2]	1.7	4.7	0.0302	0.85–1.51	mmol/L
	Glutamyl transpeptidase	41.7% (20/48)	42.5 [25.8–102.8]	22.0% (33/150)	26.0 [15.2–52.5]	1.9	6.21	0.0127	<45	IU/L
	Alanine transaminase	36.7% (18/49)	36.0 [24.0–77.0]	15.3% (23/150)	19.5 [12.0–35.0]	2.4	9.07	0.0026	< 40	IU/L
	White Blood Cell Ratio	34.7% (17/49)	1.3 [1.1–1.6]	18.7% (28/150)	1.5 [1.3–1.8]	1.86	4.54	0.033	1.20–2.40	N/A

* There are no units for a part of the tests, as the result is a ratio, or is a qualitative result, or due to the definition of the result. The unit field of these tests was recorded as N/A (not available). Abbreviation: AHI, acute HIV infection; HIV-, HIV-negative group; LR+, positive likelihood ratio.

## Data Availability

A example dataset of laboratory tests results was provided for code validation (see Appendix A). Further inquiries can be directed to the corresponding author.

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
