# Peer review of "A More Comprehensive Clinical and Laboratory Characterization of 61 Acute HIV Infection Patients in Southwest China"

_pathogens, 2023, doi:10.3390/pathogens12010142_

Round 1
Reviewer 1 Report
Please see attached file.

Author Response
Dear Reviewer,
Thank you very much for your valuable comments on our paper. We have studied your valuable comments and tried our best to revise the manuscript. Please see the attachment for point-to-point responses
We apologize for submitting the revised version so late. Based on your comments, we have supplemented the control group data and its workload exceeded our expectation. Thank you again for giving your valuable time to help us improve the quality of the manuscript.
Thank you again for your comments and suggestions.
Sincerely yours,
Prof. Chuanmin Tao
(taocm@scu.edu.cn)

Reviewer 2 Report
GLOBALY
=> The data are well presented but the facts are already known and do not bring some more information.
=> Epidemiological data should be compared to data from other stages of HIV diagnosis, especially in China. If no study describes AHI in China, this should be the angle through with the date should be reported.
=> Do the authors have data on the “abnormal results” in non-HIV patients in each of the department HIV patients are diagnosed. At least data from the literature? It would be useful do see which associations of markers are more prone to trigger a HIV-test.
=> Overall, the article need major change in the way the results are discussed.
SUMMARY
L9 : « compromise », misuse of the word. Change for a term like “failure”
L9: “were missed” => “are missed”
L13: “programmatically » => unclear. reformulate
INTRODUCTION
Suggestion: use present instead of preterit throughout the introduction:
=> L30 “is responsible, L37 ‘is massively integrated”, L40 “are attributed”…
L30: “and was responsible for 37.7 (30.2–45.1) million infections and 680 000 [480 000– 30 1.0 million] deaths in 2020 [1]”. Reformulate
L35: “to with a P31 band[2]. » => « to POSITIVE with a P31 band”
L44: ref 9 is outdated and target a specific geographic area. Any ref for China ?
L49: AHI is known since long and many studies described it.
=> The author should mention the specific situation in China regarding HIV, AHI, and maybe address specific questions regarding risk groups.
MAT AND METH
L56: “was most widely used” => “is most widely used”
L71: “since they were not reported to CDC in 71 our hospital” => unclear, reformulate
L97 (and subsequently): “abnormality rate all laboratory tests” => change for a term like “rate of out-of-range laboratory results”
L98: “LIS » => define (Lab information system ?)
L103: “before this medical visit” => which medical visit ? Reformulate and precise.
L105: ” For that reason, we calculated the time difference between each record and the first positive HIV screening record of the corresponding patient, a record was only included into the subsequent analysis if its absolute value of the time difference was less than or equal to 3.
=> split into 2 sentences.
=> what is the unit here ? 3 months ?
RESULTS
Throughout the results, figures and discussion, replace “homosexual” by “men having sex with men” (MSM = consecrated term that should be used), or “sex between men”, according to the context.
This term encompasses all men having sexual intercourses with other men, without judging their sexual orientation.
Do you have data on other STIs in these AHI patients ?
Fig1B:
ED => change for ER
L169/171: “combined result » => change for « combo test results » (and subsequently, in the discussion”
Fig 2A: relace X2 by H
Fig 2D: describe the blox plots, and the circles above them
L201: “Abnormality rate of all laboratory test”
L209: ”these tests with high abnormality rates fall into six main groups” : specify in the table to which group belongs each marker.
DISCUSSION
L218” In recent years, the mode of HIV transmission has changed widely, and the propor-218 tion of sexually transmitted infections has increased greatly in China”
=> in recent years : can you be more precise ?
=> can you precise in which way it changed? HIV is mostly sexually transmitted in Europe and the Americas (with vertical transmission, and parenteral transmission being secondary) but some strains can diffuse specifically in persons who inject drugs (PWID) like seen in south-east Asia. Somme more context would be welcome.
L229-231: “this implies that patients at this age may have a low probability of medical visit during AHI, or may be easy to be neglected during the visit.
=> or other explanations like higher rates of sexual risk behaviour in young age / or the recoursing to sex workers in older men ? There could be cofounders with the type of sexual relationships and age. To be discussed.
L244-247: ” In contrast, the dyspnea-related symptoms observed in this study have rarely been reported previously, except in an earlier prospective study[19] and several case reports[11,20,21], this may be because descriptions of these symptoms were too diverse to be realized that they can be summarised, although these descriptions share similar meanings and rarely occur alone in one patient.”
=> too long. Split the sentence in two, and reformulate (“to be realized” = incorrect)
=> are these results consistent over the whole period? 2020-2021 correspond to the SARS-CoV-2 pandemic ? Did AHI patients receive COVID testing? That could also explain the dyspnea-related symptoms.
L262: “Theoretically, the detectability of HIV P24 Ag and HIV Ab are mutually exclusive, because they could close each other's binding sites”;
=> It depends on the patient’s level of Ab, as well as their quality (masking recognition sites used in the assay or not). Classically, p24 can sometimes be detected in stages III, IV and later.
L268; “Although these characteristics shown by HIV Duo cannot improve the proportion of 268 AHI patients receiving screening, they do reduce loss to follow-up resulting from non-269 positive confirmatory tests”
=> this needs more explanations. the confirmatory test (WB) : is negative at Fiebig stages II and III and undetermined at Fiebig stage IV.
=> Do you rather mean that you miss less AHI with the Duo test?
=> is the difference significant?
L282; “However, a small portion of them was totally unanticipated, for instance, the conductivities of urine and urine protein, in fact, even the urine specimen itself receives little attention in HIV-related research.
=> is this really relevant for HIV diagnosis?
L285” The high abnormality results found in this study can be used to develop risk scoring algorithms to identify patients at high AHI risk. As not all patients would received these laboratory tests, laboratory-based algorithms cannot be applied to all patients like the commonly used symptom-based algorithms. Fortunately, most of the high abnormality tests in this study were from routine clinical tests, which most patients would received at the time of their visit, so the laboratory-based algorithm can still be applied to most people.
=> I do not think this is applicable in practice. Real-life patients are diagnosed in different clinical entities, where, as you said, not all the tests are performed.
=> can you devise such an algorithm (what marker should be used, and is this feasible)?
L298-305: ” In summary, […] who need to receive HIV screen-ing” => this is exactly what is written in your conclusion.
CONCLUSION
=> “AHI patients were male-dominated” => reformulate, “AHI patients were mostly males”
=> The HIV Duo assay may reduce the loss to follow-up of AHI patients thanks to 309 the downward trend of its HIV P24 Ag results, the upward trend of its HIV Ab results, 310 and its capacity to detect HIV P24 Ag from Fiebig â…¡-â…¤ stages => as already said, unclear why
=> the most common way to diagnose early HIV infection is to target risk groups (MSM, sex workers, PWID, migrants) and to propose HIV testing in patients with a mononucleosis syndrome, a diagnosis of STIs, a flu-like syndrome with history of sexual exposure, etc…
=> not sure that screening al patient with elevated PCT and D-Dimers is the most efficient way, as patients with severe bacterial infections often exhibit those.
Author Response

(The authors gave the same response as above.)

Reviewer 3 Report
Point 1: In this research article, the authors present a Comprehensive Clinical and Laboratory Characterization of Acute HIV Infection Patients; this study is highly unique and original. I recommend the following comments to the authors.
There are more grammatical errors and spelling mistakes (from abstract to conclusion), and an extensive English revision is needed.
Point 2: Did there any statistical significance in abnormality rates of six main groups such as T-cell subsets, inflammatory markers, routine urine test, blood biochemical analysis, blood routine analysis, and biomarkers of coinfecting viruses.
Point 3: Furthermore, the authors need to add one additional figure to describe the fundamental mechanisms workflow of Acute HIV infection (AHI), HIV screening assay, and patients inclusion criteria, etc. Otherwise, it would be difficult for the reader to capture the overall picture of the study.
Author Response

(The authors gave the same response as above.)

Round 2
Reviewer 1 Report
Please see attached file.

Author Response
Dear Reviewer,
Thank you very much for your valuable comments on our paper. We have studied the valuable comments from you, and tried our best to revise the manuscript. Please see the attachment for point-to-point responses.
Thank you again for giving your valuable time to help us improve the quality of the manuscript.
Sincerely yours,
Prof. Chuanmin Tao
(taocm@scu.edu.cn)

Reviewer 2 Report
Thanks for answering all my majors and minors concerns.
The quality of the paper is much greater.
Author Response
Dear Reviewer,
Thank you for your approval of our paper. The point-to-point responds are listed as follows in a different color.
Thank you again for giving your valuable time to help us improve the quality of the manuscript.
Sincerely yours,
Prof. Chuanmin Tao
(taocm@scu.edu.cn)
Point 1: Thanks for answering all my majors and minors concerns. The quality of the paper is much greater.
Response 1: Thank you for your approval, and thank you again for your valuable and highly practical suggestions!
Round 3
Reviewer 1 Report
It has to be specified that Supplemental Table 2 should be opened in Notepad or EmEditor.
The authors should add their response on Point 2 to the "Conclusions".
Author Response
Dear Reviewer,
Thank you very much for your valuable comments on our paper. We have studied the valuable comments from you. The point-to-point responds are listed as follows in a different color.
Thank you again for giving your valuable time to help us improve the quality of the manuscript.
Sincerely yours,
Prof. Chuanmin Tao
(taocm@scu.edu.cn)
Point 1: It has to be specified that Supplemental Table 2 should be opened in Notepad or EmEditor.
Response 1:
Thank you for your comment, we have added a paragraph to the "supplementary material" section to clarify this issue, and the paragraph is as follows:
It should be noted that we provide two Dataset S1, the two files are in different formats but with exactly the same content. The xlsx file can be read using excel, while the csv file is more suitable for programming, and can be opened using Notepad or EmEditor (https://www.emeditor.com/).
Point 2: The authors should add their response on Point 2 to the "Conclusions".
Response 2:
Thanks to your suggestion, we have added a paragraph at the end of the "Conclusions" to explain this issue:
Building a risk-scoring algorithm based on these tests may allow for better identification of AHI, but an algorithm built on prospective data may be more valuable, so a specific algorithm was not built in this study, and it is hoped that future prospective studies will give more attention to these tests.